# Associated Factors and Immune Response to the Hepatitis B Vaccine with a Standard Schedule: A Prospective Study of People with HIV in China

**DOI:** 10.3390/vaccines11050921

**Published:** 2023-04-29

**Authors:** Li Nie, Wei Hua, Xiuying Liu, Xinghuo Pang, Caiping Guo, Wei Zhang, Yakun Tian, Qian Qiu

**Affiliations:** 1Beijing Center for Disease Prevention and Control, Beijing 100013, China; niel@bjcdc.org (L.N.);; 2Clinical and Research Center for Infectious Diseases, Beijing Youan Hospital, Capital Medical University, Beijing 100069, China

**Keywords:** hepatitis B vaccine, HIV/AIDS, immune response, associated factors, China

## Abstract

Hepatitis B (HB) vaccination is recommended for people with human immunodeficiency virus/acquired immune deficiency syndrome (HIV/AIDS). We aimed to assess the immune response to the HB vaccine and associated factors using the standard vaccination schedule among people with HIV (PWH) in China. A prospective study was carried out from 2016 to 2020 in Beijing, China. PWH were given three 20 μg doses of recombinant HB vaccine at 0, 1, and 6 months. Blood samples were taken within 4–6 weeks after each dose to evaluate the anti-HBs levels. A total of 312 participants completed vaccination and serologic testing. The seroconversion (anti-HBs ≥ 10 IU/L) rates following the first, second, and third doses of the vaccine were 35.6% (95% CI: 30.3–40.9%), 55.1% (95% CI: 49.6–60.7%), and 86.5% (95% CI: 82.8–90.3%), respectively, and the geometric means of the anti-HBs titers were 0.8 IU/L (95% CI: 0.5–1.6 IU/L), 15.7 IU/L (95% CI: 9.4–26.3 IU/L), and 241.0 IU/L (95% CI: 170.3–341.1 IU/L), respectively. In multivariate analysis, after three doses of vaccination, age, CD4 cell count, and HIV-RNA viral load were significantly associated with strong, moderate, and weak response, respectively. These findings confirm that these personal health conditions are related to the HB response. HB vaccination in PWH using the standard schedule was still highly effective in the context of early treatment initiation, especially among participants aged 30 years and younger.

## 1. Introduction

Historically, hepatitis B virus (HBV) infection has been a major public health concern in the Asia–Pacific region, particularly in China [1,2]. People with HIV/AIDS (PWH) face a high risk of HBV infection since these viruses have shared modes of transmission. HBV prevalence among PWH is approximately 8–19% higher than that among the general adult population in China [3,4,5,6,7]. Due to their compromised immune systems, PWH are more likely to experience fast disease progression and severe symptoms after exposure to HBV [8]. Therefore, effective prevention and control of HBV is essential for PWH.

HB vaccination is recommended in PWH as the most important HB prevention method. Numerous studies have been carried out to determine an optimal vaccination schedule, and the seroconversion rates for HB vaccines in these studies varied widely depending on the schedule and number of doses, ranging from 34% to 88.6% [9,10,11]. The effectiveness of HB vaccines in PWH in China has rarely been reported, although a national Expanded Program on Immunization (EPI) has been carried out among infants since 1992. A systematic review has suggested that higher doses or prolongation of the vaccination schedule may improve vaccine efficacy [12], but there is no consensus on the best HB vaccination schedule among PWH. A recent study in China showed that a high dosage of the HB vaccine (60 μg) had almost the same efficacy as the standard dosage (20 μg) 4 weeks after completion of the full course of treatment (three doses) [13]. Moreover, the standard schedule is still the most accessible and affordable HB vaccination schedule in China. PWH are relatively invisible and mobile, and HB vaccination and administration for migrants is challenging. For this reason, PWH are in great need of a convenient and easy vaccination program [14]. The effectiveness of the standard schedule is therefore critical for HIV-infected Chinese adults.

Furthermore, the seroconversion rate of the HB vaccine may be associated with age, sex, CD4 cell count, CD4/CD8 ratio, and HIV-RNA viral load in PWH [9]. These factors may help patients and doctors choose the best time to deliver the HB vaccine. Meanwhile, following recent developments in antiretroviral therapy (ART) with new drugs coming into use [15,16], life expectancy for PWH has been extended and personal health circumstances have changed in China, as in other areas of the world [15]. The influence of these factors, and even the effectiveness of the HB vaccine, may vary among HIV-infected Chinese adults with different backgrounds.

For these reasons, we conducted this study to assess the immune response to HB vaccination and influencing factors using the standard vaccination schedule among HIV-infected adults in China.

## 2. Materials and Methods

### 2.1. Study Design and Participants

We carried out a prospective study of HB vaccination using the standard schedule in PWH from 2016 to 2020 in Beijing, China. The study site was Beijing Youan Hospital, a facility that specializes in treating infectious diseases and is certified to administer vaccinations for adults. This hospital receives almost 60% of the PWH treated in Beijing.

PWH aged ≥18 years with an HIV infection or AIDS qualified to take part in the study if they had never received an HB vaccine and had negative HB surface antigen (HBsAg), HB surface antibody (anti-HBs), and HB core antibody (anti-HBc) tests. Individuals with contraindications to HB vaccination were excluded. Participants were given three 20 μg doses of yeast-derived recombinant HB vaccine at 0, 1, and 6 months. Within 4–6 weeks after each dose, blood samples were taken to evaluate anti-HBs levels.

Each person was compensated with an approximately CNY 80 value travel allowance as well as one free counseling service per follow-up visit.

### 2.2. Survey and Medical History Data Collection

An in-person interview was conducted when participants received their first dose of the vaccine. Demographic and personal health data were collected, including sex, age, education, marriage status, place of birth, income, concomitant diseases (hypertension, diabetes, tuberculosis, syphilis, etc.), and related behavior. Participants were also asked to provide medical reports covering their HIV/AIDS diagnosis date, prior HIV/AIDS treatment, CD4 cell count, CD4/CD8 ratio, and HIV-RNA viral load over the last three months. Participants who did not have these reports were tested for free before administration of the first dose of the vaccine.

### 2.3. Laboratory Testing

HBsAg, anti-HBs, and anti-HBc were tested quantitatively at baseline and at each follow-up using the chemiluminescent microparticle immunoassay (the Elecsys 6000 or Modular Analytics E601 analyzer, Roche Cobas). The lower and upper detection limits for anti-HBs titer were 2 and 1000 IU/L, respectively. CD4 cell counts were tested using a four-color flow cytometry (BD Biosciences, San Jose, CA, USA). The HIV-RNA pla sma viral load was determined using the COBAS AMPLICOR HIV-1 Monitor Test (Version 1.5, Roche, Switzerland), with a lower detection limit of 40 copies/mL.

### 2.4. Statistical Analysis

The primary outcome was the seroconversion rate, defined as the proportion of respondents with anti-HBs titers of 10 IU/L or higher. Considering the lower and upper detection limits for anti-HBs titer, maximum likelihood estimation for censored, log-transformed antibody titers was used to calculate geometric mean titers (GMTs) and their 95% confidence intervals (95% CI) [17].

Descriptive statistical analysis was conducted to determine the mean (SD) or median (interquartile range) for each continuous variable and the frequency for each categorical variable. To explore the factors associated with response after each HB vaccination series, categorical factors were compared between respondents and non-respondents using the chi-square test or Fisher’s exact test. Student’s *t*-test, the Kruskal–Wallis test, or Mann–Whitney test was performed to analyze continuous variables. The Cochran–Armitage trend test was used for age-specific seroconversion rates. Multiple comparisons of GMTs between age groups and CD4 count groups were adjusted using SNK.

Polytomous logistic regression was used to analyze factors associated with four levels of immune responses in month 1, month 2, and month 7. Strong response (anti-HBs ≥ 1000 IU/L), moderate response (100 IU/L ≤ anti-HBs > 1000 IU//L), weak response (10 IU/L ≤ anti-HBs > 100 IU/L), and no response (anti-HBs **<** 10 IU/L) were the four categorical dependent variables, and base category was the group of no response. Independent variables that were significant at *p* < 0.1 in the univariate analysis were included, and those with *p* < 0.12 were retained in the polytomous logistic regression models using the stepwise method. To ensure the robustness of the models, the independent variable and intercept which are not statistically significant in logit functions were eliminated, and in such cases the parameters would be adjusted accordingly until all the models and the independent variables included had significant statistics. Adjusted odds ratios and 95% confidence intervals were estimated by the models. Statistical significance was determined at *p* < 0.05. SPSS 21.0 was used for all statistical analyses.

## 3. Results

This section provides a concise and precise description of the experimental results, their interpretation, and experimental conclusions that can be drawn.

### 3.1. Baseline Characteristics

In total, 2191 PWH were screened: their HBV markers were tested using a blood sample and they completed a screening survey about their HB vaccination history. Overall, 324 of the study’s 564 eligible patients took part. Of these, 312 HIV-infected patients received all three vaccinations and completed post-vaccination serologic testing, while 12 participants dropped out of the study (Figure 1).

Of the 312 patients, the mean age at enrollment was 34 years (ranging from 18 to 73 years); 95.8% were male, and no one identified themselves as transgender; 81% of the participants were born outside Beijing; and 58.3% had a college degree or higher degree. Nearly two-thirds of the participants had never been married. The majority (88%) of patients were men who had sex with men (MSM). The mean age at HIV diagnosis was 31.4 years. Most patients (98%) used HARRT; the median duration of HARRT was two years. Before their first vaccine dose, 81 patients (26.0%) had a CD4 cell count of less than 350 cells/mm^3^, and the median CD4 cell count was 482.8 cells/mm^3^, ranging from 30.1 cells/mm^3^ to 1338.0 cells/mm^3^. The median CD4/CD8 ratio was 0.54. More than 74% of the participants had an undetectable HIV-RNA viral load (lower than 40 copies/mL). Some participants (6.4%) had other chronic and sexually transmitted diseases. Nearly one-third of the patients had smoked in the past 6 months. The baseline characteristics of the participants are shown in Table 1.

### 3.2. Response to HB Vaccination

After the first, second, and third doses of the vaccine, the seroconversion rates were 35.6% (95% CI: 30.3–40.9%), 55.1% (95% CI: 49.6–60.7%), and 86.5% (95% CI: 82.8–90.3%), respectively, and the geometric mean titers (GMTs) of anti-HBs were 0.8 IU/L (95% CI: 0.5–1.6 IU/L), 15.7 IU/L (95% CI: 9.4–26.3 IU/L), and 241.0 IU/L (95% CI: 170.3–341.1 IU/L), respectively. As Figure 2 demonstrates, the seroconversion rates and GTMs increased sharply after the third dose. Overall, 67.3% of patients had anti-HBs titers ≥ 100 IU/L. The percentage of patients who responded strongly (anti-HBs ≥ 1000 IU/L) was relatively low, i.e., 7.4% in month 1, 15.1% in month 2, and 29.2% in month 7 after the first dose of HB vaccination (Figure 3). The seroconversion rates decreased significantly with increasing age, from 93.5% to 69.6% (*p* = 0.0001; Figure 4).

### 3.3. Factors Associated with Response to HB Vaccination

#### 3.3.1. Univariate Analysis Results

Compared with non-respondents, respondents were younger (mean age: 33.3 vs. 38.7 years), less likely to have been married (69.6% vs. 50.0%), and more highly educated (college education or higher: 61.1% vs. 40.5%) (Table 1).

Respondents were diagnosed with HIV earlier than non-respondents (mean age: 30.5 vs. 36.8 years). They had spent longer using HARRT (median: 2.0 vs. 1.0 years), had higher CD4 cell counts (median: 494.4 vs. 325.6 cells/mm^3^) and CD4/CD8 ratios (median: 0.56 vs. 0.38), and were more likely to have an HIV-RNA viral load lower than 40 copies/mL (77.8% vs. 52.4%). Age at HIV diagnosis was correlated with age at enrollment (Pearson correlation coefficient: r = 0.96, *p* < 0.01). Respondents were also less likely to have other sexually transmitted and chronic diseases (4.8% vs. 16.7%).

#### 3.3.2. Multivariate Analysis Results

In the multivariate analysis, different factors were associated with the four levels of serological response at the three time points. Age (Moderate response: AOR = 0.43, 95% CI: 0.27–0.71; Weak response: AOR = 0.38, 95% CI: 0.21–0.68) and level of education (Moderate response: AOR = 2.33, 95% CI: 1.15–4.73; Weak response: AOR = 4.28, 95% CI: 1.69–10.86) were significantly associated with the moderate and weak response to the first dose of the vaccine. Age was significantly associated with all levels of response to the second dose of the vaccine in month 2. In month 7, following the complete course of vaccination, three factors, age, HIV-RNA viral load, and CD4 cell count, were associated with all levels of response (Table 2).

#### 3.3.3. GMTs of the Anti-HBs in Different Groups

According to the multivariate analysis in month 7, the main indicators of efficacy were age, CD4 cell count, and HIV-RNA viral load. The GMTs of anti-HBs in the groups based on these factors are shown in Figure 5. Patients aged up to 30 years (GMT: 400.45 IU/L, 95% CI: 262.76–610.30 IU/L) and those aged 31–40 years (GMT: 202.94 IU/L, 95% CI: 109.27–376.89 IU/L) had significantly higher anti-HBs GMTs than those older than 40 years (GMT: 88.69 IU/L, 95% CI: 36.86–213.38 IU/L). Anti-HBs GMTs decreased with the CD4 cell count. GMT was significantly lower (GMT: 49.5 IU/L, 95% CI: 23.18–105.68 IU/L) among patients with CD4 cell counts of <350 cells/mm^3^ and those with CD4 cell counts of 350–500 cells/mm^3^ (GMT: 356.81 IU/L, 95% CI: 185.39–686.75 IU/L), slightly lower than among those with ≥500 cells/mm^3^ (GMT: 409.90 IU/L, 95% CI: 271.54–618.74 IU/L). Patients with an HIV-RNA viral load of <40 copies/mL had significantly higher anti-HBs GMTs (GMT: 311.41 IU/L, 95% CI: 213.49–454.23 IU/L) than those with an HIV-RNA viral load ≥40 copies/mL (GMT:106.57 IU/L, 95% CI: 49.93–227.46).

## 4. Discussion

This study is the first prospective analysis to explore the effectiveness of the HB vaccine and associated factors using the standard dosing schedule among HIV-infected adults in China. We found that the proportion of respondents one month after receipt of all three vaccine doses was 86.5%, showing that vaccination with 20 μg recombinant HB vaccine at 0, 1, and 6 months was effective in adults with HIV/AIDS in China. Our findings also demonstrated that the CD4 cell counts and HIV-RNA viral load had independent effects on the response after age was considered. Concomitant diseases showed a slight trend towards decreasing the proportion of respondents.

At the end of 2020, there were an estimated 1,053,000 PWH in China [18]. Among new cases reported from January to October 2021 in Beijing, 77.27% were MSM [19]. The majority of the participants in this study (MSM with an average age of 34 years) is representative of PWH in Beijing. Importantly, since 2016, most of the participants in our study had initiated ART regardless of their CD4 cell counts, reflecting the most recent international guidelines. The effectiveness of the HB vaccine in this population, therefore, represents the current real-world situation.

The standard vaccination schedule is more accessible for HIV-positive adults in China than schedules requiring higher dosages or a longer duration. This study shows that its effectiveness is slightly higher than that of the same schedule and close to that reported with a higher dosage and a longer duration. A large randomized trial evaluating three-dose 20 μg and four-dose 40 μg HB vaccination schedules in adults with HIV reported seroconversion rates of 65% and 82%, respectively [20]. Prospective studies and randomized trials have reported similar response rates of 50–62% [21,22], while other studies have reported similar or better rates, ranging from 84.15% to 92% [13,23,24]. One study, a randomized trial conducted in China, reported no significant difference in the seroconversion rate between standard and high dosages (20 μg vs. 60 μg) of recombinant HB vaccines (84.15% (69/82) vs. 90.36% (75/83)) [13]. However, these studies with higher seroconversion rates exclusively enrolled patients with CD4 counts of 200 cells/mm^3^ or higher or included patients regardless of whether they had received an HB vaccine in the past. In contrast, our study included participants with lower CD4 cell counts who had never received an HB vaccination. Our findings, therefore, reflect that currently, the standard schedule is the most accessible way to protect PWH from HB infection in China, with a higher seroconversion rate observed in younger subgroups. We also observed that more than 67% of PWH achieved a moderate or strong anti-HBs response. GMT in this study was 241.0 IU/L. Long-term persistence of response has been associated with anti-HBs titer after vaccination [25,26]. Our study showed that at least two-thirds of respondents were fully protected for a relatively long time. Given the differences in the upper detection limit and test methods, it is difficult to compare our titers directly with those reported in other studies.

Compared with healthy adults in China [27], the response to HB vaccination was diminished in PWH. This may be due to defects with T cells and B cells in HIV patients. The recombinant HB vaccine contains HBsAg, which acts as a thymus-dependent antigen. The process of this antigen triggering activation, proliferation, and differentiation of B cells into antibody-secreting plasma cells and memory B cells is T cell-dependent, requiring T cell activation and coordinated secretion of helper T cells (Th) 1 and Th2-type cell factors. However, in PWH, there may be a high proportion of Treg cells [28], impaired early T cell activation [29], inadequate cytokine secretion by helper T cells [30], depletion of resting memory B cells, and expansion of immature transitional B cells [31], which affect the overall immune response mechanisms. Meanwhile, their anti-HBs titers may decrease faster than those of healthy adults. Therefore, even when PWH have high anti-HBs titers, it is recommended to test their anti-HBs titers regularly [32].

Furthermore, this study tracked the proportion of respondents and anti-HBs GMT following each vaccine dose. After two doses, the response rate and GMT were only 55.1% and 15.7 IU/L, respectively. A third dose substantially improved the vaccine effectiveness. Therefore, in clinical practice, it is essential to complete the standard vaccination schedule; otherwise, patients with lower anti-HBs titers will not be fully protected by the vaccine.

In addition, we have reported the factors associated with the four levels of immune responses following each dose. Age at enrollment was an independent factor significantly associated with all levels of response to HB vaccination in all models, following each dose. As previous studies in PWH [20,22,33] and healthy adults [34,35] have shown, the seroconversion rate decreases as age increases. Notably, younger patients were likely to respond to the HB vaccine after one and two doses, and GMT in patients aged 30 years and younger was more than four times the GMT in patients older than 40 years. Taking age into account, the level of education was associated only with the vaccine moderate and weak response in month 1. It is possible that patients with a college degree or higher are more informed, allowing them to improve their immune state by being more aware of their health; alternatively, these patients might have been more likely to have received vaccinations as infants.

As in other studies, the baseline CD4 cell count following the second dose played a vital role in the HB vaccination response in our study [20,24,33,36,37]. Crucially, the GMT of participants with CD4 counts of 350 cells/mm^3^ or lower was 49.5 IU/L, significantly lower than that in the other two groups. This provides strong evidence for using CD4 cell count cutoffs in clinical practice. After consideration of age and the CD4 cell count, the HIV-RNA viral load was another independent factor closely associated with the HB vaccination response. This result is at odds with that reported in previous studies [13,24,38], which showed that it was challenging to identify the independent contributions of the CD4 cell count and HIV-RNA viral load due to insufficient sample sizes in different categories. All of the factors found in our study could be explained by immunological exhaustion and immune senescence, two signaling pathways that can affect memory B cells and regulatory T cells to cause their malfunction [39].

We also observed that PWH with concomitant diseases tended to have a slightly lower seroconversion rate. Previous studies have shown similar results in the general adult population [40,41,42], but not in PWH. A meta-analysis evaluating the HB vaccine in healthy adults showed that concomitant disease can significantly reduce the immune response to HB vaccination [30]. PWH can achieve a higher life expectancy with HARRT and may therefore develop more concomitant chronic and other infectious diseases. Concomitant diseases will place a greater burden on their immune system. This factor should be considered in the clinical practice of vaccination.

There are some limitations of our study. First, we evaluated the effectiveness of the standard HB vaccination schedule in HIV-positive adults without a healthy adult control group. However, the vaccine’s effectiveness among healthy people provides a good reference. Second, relatively few female participants were enrolled in this study, which is consistent with current trends in new HIV/AIDS cases. To our knowledge, females always respond more frequently than males due to sex hormone differences, specifically due to androgen and estrogen. Third, the lifetime vaccination history was investigated through in-person interviews and may have been influenced by memory bias. In our study, 7.4% of the patients responded strongly following their first dose of vaccination, showing that these participants were more likely to have received vaccinations in the past.

## 5. Conclusions

The CD4 cell counts, HIV-RNA viral load, and age were associated with seroconversion. Taking different personal health conditions into account, the standard HB vaccination schedule remains highly effective among HIV-positive adults in China.

## Figures and Tables

**Figure 1 vaccines-11-00921-f001:**
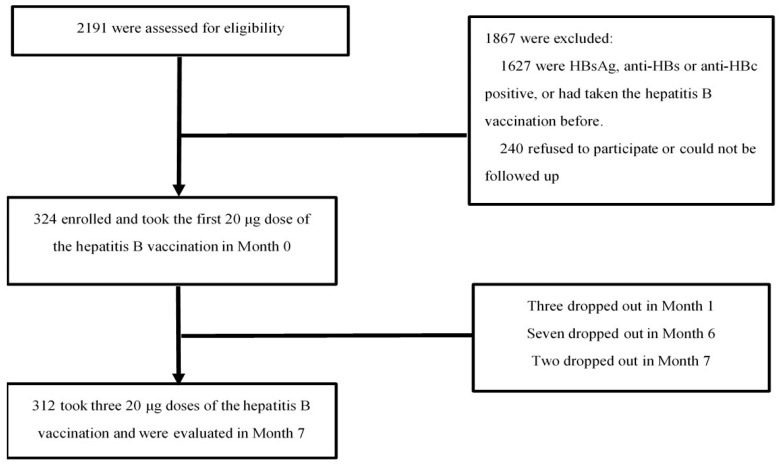
Flow chart of the study population.

**Figure 2 vaccines-11-00921-f002:**
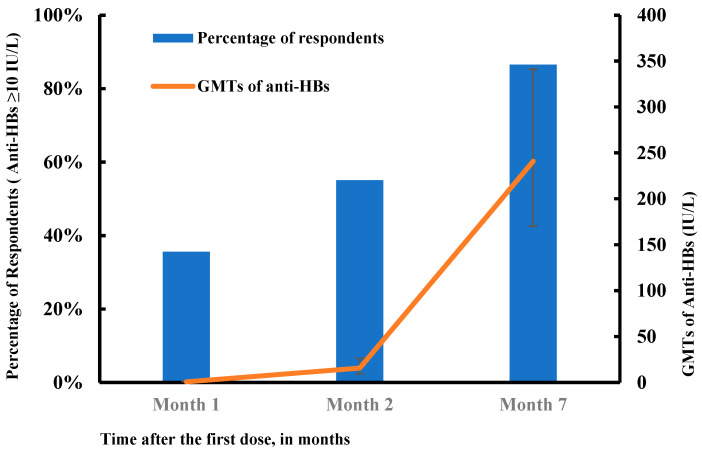
Percentage of respondents and geometric mean titers (GMTs) of anti-HBs following hepatitis B vaccination (N = 312). The error bars show the 95% confidence intervals (CIs).

**Figure 3 vaccines-11-00921-f003:**
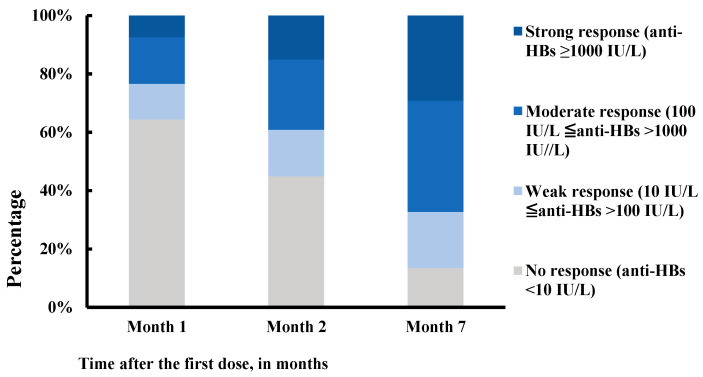
Percentage of four levels of response following hepatitis B vaccination (N = 312).

**Figure 4 vaccines-11-00921-f004:**
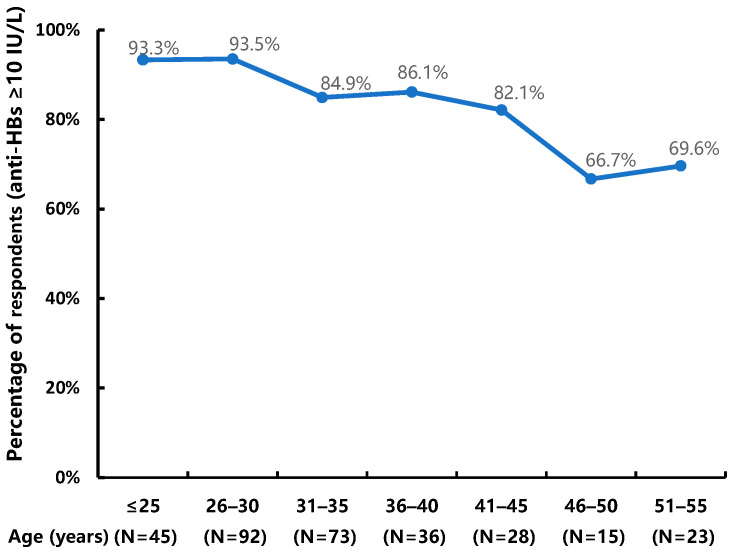
Age-specific percentages of respondents to the hepatitis B vaccine in HIV-infected adults.

**Figure 5 vaccines-11-00921-f005:**
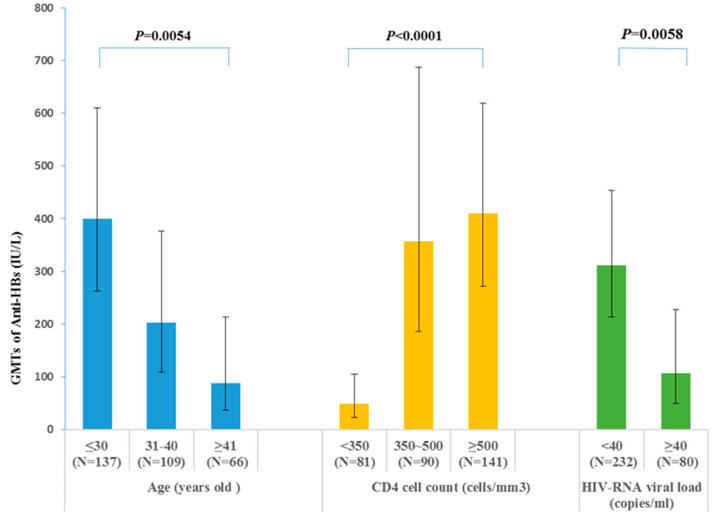
Geometric mean titers (GMTs) of anti-HBs following hepatitis B vaccination in different groups in month 7. Error bars show the 95% confidence intervals (CIs).

**Table 1 vaccines-11-00921-t001:** Demographics, HIV history, and clinical characteristics of respondents and non-respondents.

Characteristic	Total Sample (N = 312)	Respondents (N = 270)	Non-Respondents (N = 42)
**Age at enrollment, years ^†^****	34.0 (9.8) (18–73)	33.3 (9.3) (18–73)	38.7 (11.1) (18–61)
**Sex, N (%)**			
Male	299 (95.8)	259 (95.9)	40 (95.2)
Female	13 (4.2)	11 (4.1)	2 (4.8)
Transgender	0 (0)	0 (0)	0 (0)
**Place of birth, N (%)**			
Beijing	58 (18.6)	47 (17.4)	11(26.2)
Other provinces	254 (81.4)	223 (82.6)	31 (73.8)
**Marriage, N (%) ***			
Never married	209 (67.0)	188 (69.6)	21 (50.0)
Married	103 (33.0)	82 (30.4)	21 (50.0)
**Education, N (%) ****			
High school diploma or less	130 (41.7)	105 (38.9)	25 (59.5)
College degree or higher	182 (58.3)	165 (61.1)	17 (40.5)
**Transmission route, N (%)**			
MSM	275 (88.1)	238 (88.2)	37 (88.1)
MSWM and others	37 (11.9)	32 (11.8)	5 (11.8)
**Occupation, N (%)**			
Fixed	265 (84.9)	231 (85.6)	34 (81.0)
Non-fixed	47 (15.1)	39 (14.4)	8 (19.1)
**BMI ^ †^**	22.2 (3.2) (15.7–36.3)	22.2 (3.2) (15.2–36.3)	22.6 (3.2) (18.0–33.1)
**Age at HIV diagnosis, years ^†^****	31.4 (9.7) (7–68)	30.5 (9.0) (17–68)	36.8 (12.4) (7–60)
**Time since HIV diagnosis, years ^§^****	2.0 (4.0) (0–17)	2.0 (3.0) (0–17)	1.0 (2.0) (0–17)
**Time using ART, years ^§^****	2.0 (3.0) (0–17)	2.0 (3.0) (0–17)	1.0 (1.0) (0–17)
**CD4 cell count at enrollment, cells/mm** ^ **3 §** ^ ******	482.8 (308) (30.1–1338.0)	494.4 (276.7) (40.7–1338.0)	325.6 (305.5) (30.1–851.3)
**CD4/CD8 ratio ^ §^****	0.54 (0.46) (0.02–2.64)	0.56 (0.44) (0.02–2.64)	0.38 (0.45) (0.05–1.11)
**HIV-RNA viral load < 40 copies/mL, N (%) ****	232(74.4)	210 (77.8)	22 (52.4)
**Concomitant diseases, N (%) ****	20 (6.4)	13 (4.8)	7 (16.7)
**Smoked in the past 6 months, N (%)**	100 (32.1)	84 (31.1)	16 (38.1)

Abbreviations: MSM, men who have sex with men; MSWM, men who have sex with men and women; BMI, body mass index; ART, antiretroviral therapy. ^†^ Mean (SD) (range); ^§^ Median (interquartile range) (range); * *p* < 0.05; ** *p* < 0.01.

**Table 2 vaccines-11-00921-t002:** Multivariate analysis of factors associated with the four-level response to vaccination at three time points (base category: no response, anti-HBs < 10 IU/L).

Time Points	Response	Characteristic	Multivariate OR (95% CI)	*p* Value
Month 1	Strong response	**Age (years)**		
		≤30	Reference	
		31–40, ≥41	0.46 (0.25–0.87)	0.0162
	Moderate response	**Age (years)**		
		≤30	Reference	
		31–40, ≥41	0.43 (0.27–0.71)	0.0007
		**Education**		
		High school diploma or less	Reference	
		College degree or higher	2.33 (1.15–4.73)	0.0192
	Weak response	**Age (years)**		
		≤30	Reference	
		31–40, ≥41	0.38 (0.21–0.68)	0.0012
		**Education**		
		High school diploma or less	Reference	
		College degree or higher	4.28 (1.69–10.86)	0.0022
Month 2	Strong response	**Age (years)**		
		≤30	Reference	
		31–40, ≥41	0.33 (0.20–0.54)	<0.0001
	Moderate response	**Age (years)**		
		≤30	Reference	
		31–40, ≥41	0.31 (0.21–0.48)	<0.0001
	Weak response	**Age (years)**		
		≤30	Reference	
		31–40, ≥41	0.51 (0.33–0.79)	0.0024
Month 7	Strong response	**Age (years)**		
		≤30	Reference	
		31–40, ≥41	0.41 (0.24–0.69)	0.0007
		**HIV-RNA viral load (copies/mL)**		
		<40	Reference	
		≥40	0.35 (0.15–0.83)	0.0170
		**CD4 cell count (cells/mm** ^ **3** ^ **)**		
		<350	Reference	
		350~500, ≥500	2.54 (1.54–4.19)	0.0003
	Moderate response	**Age (years)**		
		≤30	Reference	
		31–40, ≥41	0.42 (0.26–0.69)	0.0006
		**HIV-RNA viral load (copies/mL)**		
		<40	Reference	
		≥40	0.37 (0.17–0.82)	0.0150
		**CD4 cell count (cells/mm** ^ **3** ^ **)**		
		<350	Reference	
		350~500, ≥500	2.11 (1.31–3.39)	0.0020
	Weak response	**Age (years)**		
		≤30	Reference	
		31–40, ≥41	0.54 (0.32–0.92)	0.0241
		**HIV-RNA viral load (copies/mL)**		
		<40	Reference	
		≥40	0.37 (0.15–0.90)	0.0275
		**CD4 cell count (cells/mm** ^ **3** ^ **)**		
		<350	Reference	
		350–500, ≥500	1.70 (1.02–2.84)	0.0436

Strong response: anti-HBs ≥ 1000 IU/L; Moderate response: 100 IU/L ≤ anti-HBs > 1000 IU//L; Weak response: 10 IU/L ≤ anti-HBs > 100 IU/L.

## Data Availability

The de-identified data are available upon reasonable request from the corresponding author.

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
