# Peer review of "Associated Factors and Immune Response to the Hepatitis B Vaccine with a Standard Schedule: A Prospective Study of People with HIV in China"

_vaccines, 2023, doi:10.3390/vaccines11050921_

Round 1

Reviewer 1 Report

1) This is a pretty straight forward manuscript, and so is the interpretation of the data. It appears that a subset of the patient population has one or more of the following: weakened immune system, low CD4 counts, low CD4:CD8 ratio, detectable HIV viral loads. In addition, the response to vaccination is also heterogeneous. While the multivariate analysis (Table 2) points to various factors associated with response to vaccination, it is not clear what led to a strong response. Can the Table 2 be revised to include (strong, moderate, weak, and no response patients) rather than (responders and non responders)?

2) Sexual behavior seems irrelevant in this study. Respondent and non-respondents have identical percentage splits for the two sexual population groups. Having said that, almost the entire cohort is comprised of men who have sex with men (MSM), so the study is not powered to factor sexual behavior and response to vaccination. 

Minor Comments

1) People with HIV (PWH) is preferred terminology

2) Sexual behavior descriptors are vague. Utilize SOGI guidelines. Example, Men who have sex with Men (MSM), Men who have sex with men and women (MSWM) etc instead of homosexual and heterosexual behavior.

3).  Demographics are listed as binary, and shouldn't be! Where were the transgender patients in this cohort? Perhaps the population was not queried appropriately. 

4). Regulatory approval details are not included in a study involving human subjects.

5). There is no mention of whether participants were compensated.

Reviewer 2 Report

The main question addressed by the research is, do hiv positive patients respond differently to standard doses of hep b vaccinesThe conclusions are consistent. They address the issues raised.

This is a well conducted study with good english, and good analysis, it is of average originality.

It is inclusive of those with higher and lower cd4 counts and with higher and lower viral loads to hiv.  therefore it is more real world than other studies material.

I suggested they make some comments about 'optimal vaccine efficacy of 95% in normal populations, and comment on this in the setting of hiv positive individuals

the ideal hep b vaccine will give 95 percent efficacy after three doses at schedule of 0 1 6 months.  would be good to mention why hiv patients do not reach that target.  

Author Response

We sincerely thank the reviewer for the valuable feedback. The reviewers’ comments are laid out below in italicized font and specific concerns have been numbered. Please see our respective responses (marked in red) and changes/additions to the manuscript (marked in blue).

Point 1: I suggested they make some comments about 'optimal vaccine efficacy of 95% in normal populations, and comment on this in the setting of hiv positive individuals. The ideal hep b vaccine will give 95 percent efficacy after three doses at schedule of 0 1 6 months. Would be good to mention why hiv patients do not reach that target.

Response 1: We think this is an excellent suggestion. A 95% efficacy is an ideal situation for all researchers and public health professionals, but no studies have reported such results until now. We strongly agree that further discussion of the possible causes and mechanisms would allow the readers to better understand our study.

Therefore, we have added the paragraph in discussion on page 12, lines 261-271, as follows,

” Compared with healthy adults in China [27], the response to HB vaccination was diminished in PWH. This may be due to defects with T cells and B cells in HIV patients. The recombinant HB vaccine contains HBsAg, which acts as a thymus-dependent antigen. The process of this antigen triggering activation, proliferation, and differentiation of B cells into antibody-secreting plasma cells and memory B cells is T cell-dependent, requiring T cell activation and coordinated secretion of helper T cells (Th) 1 and Th2-type cell factors. However, in PWH, there may be a high proportion of Treg cells [28], impaired early T cell activation [29], inadequate cytokine secretion by helper T cells [30], depletion of resting memory B cells, and expansion of immature transitional B cells [31], which affect the overall immune response mechanisms. ”

Reviewer 3 Report

This paper is very clear, methodology is sound and it leads with an important issue in people with HIV.

Just few comments.

I see from patient's profile that many patients were from other province. Was there any migrant patient? 

Studies from Vargas et al, 2021 demonstrated how increasing the standard dosage of HBV vaccine could improve antibody response. What about people who did not respond? Were they referred for another cycle of vaccination?

Please expand discussion by using Infez Med2021 Jun 1;29(2):236-241.

Outcome of HBV screening and vaccination in a migrant population in southern Italy

Kind regards
